# CLIP-Nav: Using CLIP for Zero-Shot Vision-and-Language Navigation

**Vishnu Sashank Dorbala**
University of Maryland, College Park

**Gunnar Sigurdsson**
Amazon Alexa AI

**Robinson Piramuthu**
Amazon Alexa AI

**Jesse Thomason**
Amazon Alexa AI

**Gaurav S. Sukhatme**
Amazon Alexa AI

## Abstract

Household environments are visually diverse. Embodied agents performing *Vision-and-Language Navigation (VLN) in the wild* must be able to handle this diversity, while also following arbitrary language instructions. Recently, Vision-Language models like CLIP have shown great performance on the task of zero-shot object recognition. In this work, we ask if these models are also capable of zero-shot *language grounding*. In particular, we utilize CLIP to tackle the novel problem of *zero-shot VLN* using natural language referring expressions that describe target objects, in contrast to past work that used simple language templates describing object classes. We examine CLIP's capability in making sequential navigational decisions without any dataset-specific finetuning, and study how it influences the path that an agent takes. Our results on the coarse-grained instruction following task of *REVERIE* demonstrate the navigational capability of *CLIP*, surpassing the supervised baseline in terms of both success rate (SR) and success weighted by path length (SPL). More importantly, we quantitatively show that our CLIP-based zero-shot approach generalizes better to show consistent performance across environments when compared to SOTA, fully supervised learning approaches when evaluated via *Relative Change in Success (RCS)*.

## 1 Introduction

*Vision-and-Language Navigation (VLN)* requires agents to follow human instructions in unseen environments. This is a challenging problem since instructions heavily rely on the contents of the scene, possibly unknown to a general agent. A common approach to VLN is to use supervised learning; we argue that this is not practical, given the drastic shift in semantics from scene to scene that impacts the performance of a trained model. We observe this phenomenon in SOTA supervised learning approaches for VLN, which have large performance drops on "unseen" environments that are absent from the training dataset.

In this work, we seek to address this issue of generalizing to new environments, and propose to solve VLN in a fully *zero-shot* manner. The agent is assumed to have no prior knowledge about the instructions or the environment. This setting is does not rely on a particular VLN dataset and is free from any environmental bias that datasets may inherently have.

A substantial body of prior work tackles VLN using supervised learning Gu et al. (2022); Guhur et al. (2021); Wang et al. (2019b); Hong et al. (2021). Models are first trained on a set of "seen" environments and instructions, before being evaluated on both "seen" and "unseen" test data. This paradigm often involves encoding instructions and learning topological relations during training Chen et al. (2021); Wang et al. (2021); Chen et al. (2022b), and imitating navigational decisions during inference. These approaches usually see a significant drop in performance on unseen data. Figure 1 compares results of popular VLN approaches on the coarse-grained instruction following task of REVERIE Qi et al. (2020). Observe the significant drop in performance on the "Unseen Val" and "Unseen Test" sets, when compared with "Seen Val" which validates on previously seen environments. This drop holds across all approaches. Performance in comparison with humans is also significantly worse, not just for REVERIE, but also for other popular instruction following tasks such as R2R Anderson et al. (2018) and SOON Zhu et al. (2021).

Current VLN models are also dataset specific; a trained model from one will not generalize to another. For instance, training on REVERIE and testing on SOON may not give comparable results, despite both involving the coarse-grained instruction following task. A critical factor affecting this

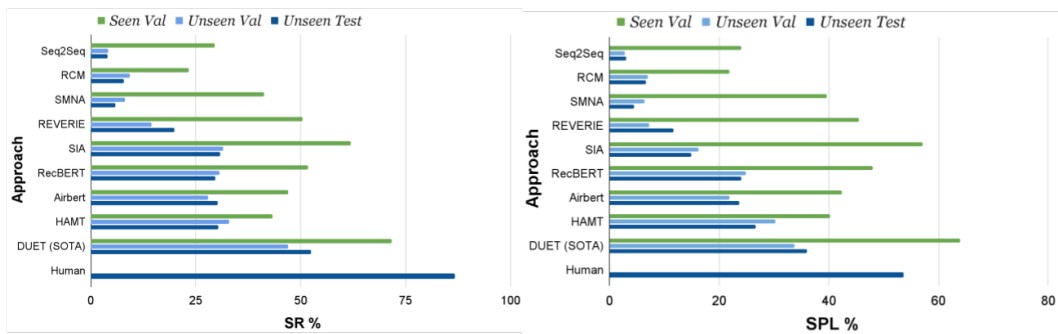

Figure 1: **Comparing Model Performance on REVERIE Seen and Unseen Splits**: Observe the significant drop in performance of the *Unseen Val* and *Unseen Test* sets (blue) when compared with the *Val Seen* set (green), across all approaches as measured by both Success Rate (SR) (*Left*) and Success Weighted by Path Length (SPL) (*Right*). Also observe the poor *Unseen Test* set performance when compared to the human baseline.

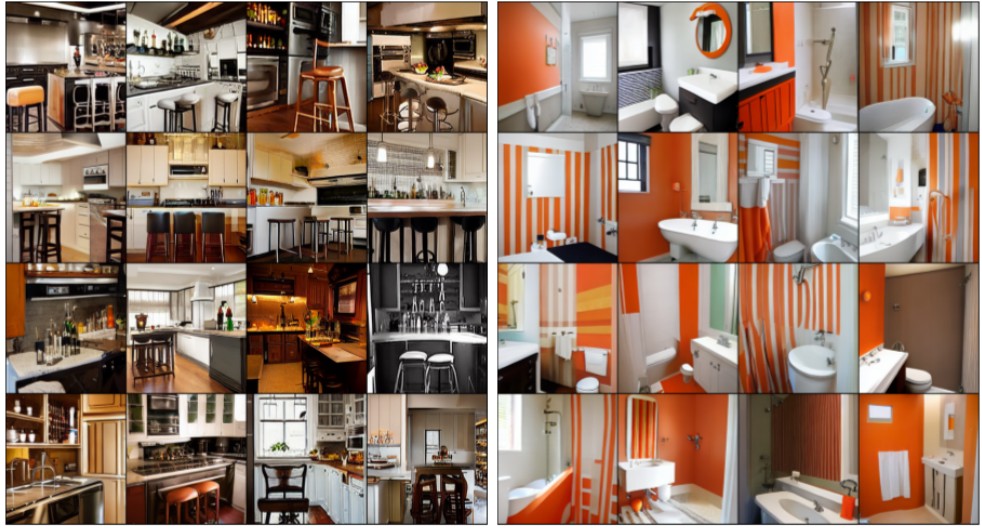

Bar chair, in the kitchen, by-the-oven.     Bathroom with orange stripes.

Figure 2: Household images generated from text using a latent diffusion model Rombach et al. (2022). Observe the variance in layout, positioning and lighting. Each home is visually unique, and we hypothesize that this causes people to use unique, environment-specific language in giving out instructions (*Orange-striped bathroom* for instance). This hypothesis is substantiated by our inspection of REVERIE, and motivates us to treat VLN as a fully *zero-shot* problem.

generalization is the lack of sufficient training data, and a few articles address this issue by proposing data augmentation techniques Chen et al. (2022c); Li et al. (2022). However, data augmentation has only shown minor improvements in performance.

Household environments are visually diverse, each with unique layouts and objects. As such, instructions given by humans to describe target locations usually contain environment-specific cues. For example, the same target item "bottle" could be described as "next to the mirror" in one instruction and as "above the sink" in another. There could also be unique environment-specific identifiers such as "*orange-striped* bathroom" or "painting depicting *crudely-drawn people*".

We hypothesize that the visual distinctiveness of environments causes people to use *unique* language involving those scenes. In our inspection of the REVERIE dataset we noticed a low pairwise average cosine similarity of $0.32$ between all seen and unseen instructions, substantiating this claim. As all environments are potentially distinct from one another, training datasets fail to capture all the visuo-linguistic diversity needed to generalize (Figure 2).

Our work makes the following contributions:

- We present a novel approach to solve coarse-grained instruction following tasks: breaking down guidance language into keyphrases, visually grounding them, and using the grounding scores to drive **CLIP-Nav**, a novel *"zero-shot"* navigation scheme.

- Further, we incorporate *backtracking* into our scheme, and present **Seq CLIP-Nav** which drastically improves CLIP-Nav's results, showcasing the importance of backtracking in solving such tasks.

- Our results establish a ***zero-shot baseline*** on the task of REVERIE Qi et al. (2020), surpassing the *unseen* supervised baseline without any form of dataset-specific finetuning in terms of SR and SPL. Our SPL results on this dataset also improves on the SOTA.

- Finally, we establish a new metric to measure generalizability in VLN tasks — **R**elative **C**hange in **S**uccess (**RCS**). This metric quantitatively showcases the improved performance of our CLIP-based approaches over other supervised methods.

## 2 RELATED WORK

**Vision-and-Language Navigation (VLN)**: Coarse-grained Qi et al. (2020); Zhu et al. (2021) and fine-grained Anderson et al. (2018); Ku et al. (2020); Vasudevan et al. (2021) instruction following tasks have recently been of great interest. A majority of approaches attempting to solve these tasks use some form of supervision, employing behavior cloning Huang et al. (2019), reinforcement learning Wang et al. (2018), or even imitation learning Wang et al. (2019a). These approaches are limited to solving VLN either on a single or a specific set of datasets, and do little to analyze cross-dataset performance.

**Zero-Shot VLN (ZSVLN)**: A recent paradigm shift in machine learning has led to the emergence of large deep learning models that are pre-trained on vast amounts of unlabeled data, and finetuned on a variety of downstream tasks. Examples of this include BERT Devlin et al. (2018) and GPT-3 Brown et al. (2020), which have shown SOTA performance on various natural language tasks. We are particularly interested utilizing CLIP Radford et al. (2021), which has shown improved zero-shot performance on several downstream vision-language tasks Shen et al. (2021). Very recently, CLIP been used to encode semantics of the environment to indirectly improve downstream zero-shot VLN performance in Shafiullah et al. (2022); Shah et al. (2022); Chen et al. (2022a). These approaches however significantly differ from our work which directly uses CLIP embeddings to make sequential navigational decisions.

Gadre et al. (2022) in *CLIP on Wheels (CoW)* attempt to utilize CLIP to perform zero-shot object navigation. Object navigation involves exploring the environment to find a target object without any *guidance instructions*. CoW modularizes this task into exploration and object localization, and uses CLIP for the latter.

While underlying task of navigating to a target location is the same, VLN is very different from object navigation. VLN uses natural human instruction language as opposed to template language. For example, a coarse-grained VLN instruction would be "Go to the kitchen on your right and water the plant there.", while object navigation uses only the target word "plant". CoW augments these words with templates (a photo of a <object> in a video game) and then grounds them with CLIP. We instead use CLIP not just to ground keyphrases extracted from the VLN instruction, but also to drive our exploration policy.

To the best of our knowledge, there are no works that attempt to solve coarse-grained instruction following in a generalized, zero-shot setting, which forms the basis of our research. We look at solving VLN without any form of dataset-specific finetuning, and seek to transfer CLIP's powerful language grounding capabilities into a sequential decision making pipeline for navigation.

## 3 APPROACH

Performing zero-shot VLN in a household environment requires the agent to have a sense of structural priors to make sequential decisions about how to get to an unseen target location. Consider Figure 3 for instance. The command is for the robot to "Go to the kitchen", but the panoramic view does not contain visual elements associated with a kitchen. In such a case, we require the agent to pick the best view (and consequently the best direction) that would potentially lead to a kitchen. In this work, we use CLIP to make sequential navigational decisions, asserting its capability in capturing structural priors of indoor hosuehold environments.

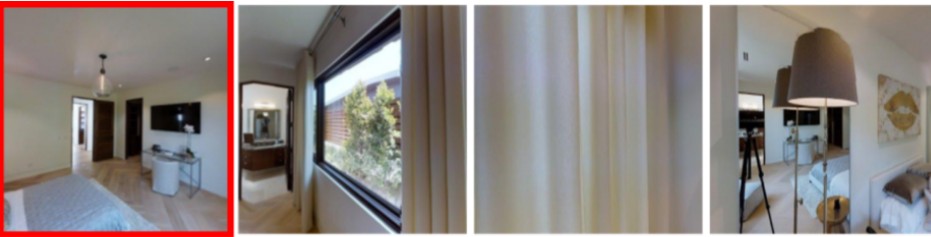

**Instruction - "Go to the kitchen"**

Figure 3: We look at CLIP's ability to make sequential navigational decisions. Here, the instruction "*Go to the kitchen*" suggests that the agent needs to leave the room. However, in order for it to make this decision, it needs to ground this instruction within the panorama, to choose a view with the door leading outwards. Notice that there are no clear visual entities (i.e. *spoons* or *sinks*) to suggest the chosen image (in red) is a "*kitchen*". The decision is based on pretrained CLIP's structural prior of the household in picking a view that might lead to the kitchen.

Our approach consists of three steps:

1. **Instruction Breakdown** - Decompose coarse-grained instructions into keyphrases.
2. **Vision-Language Grounding** - Ground keyphrases in the environment using CLIP.
3. **Zero-Shot Navigation** - Utilize the CLIP scores to make navigational decisions.

Given that our approach is zero-shot and requires no finetuning, we could choose any fixed set of datapoints for evaluation. We opt to use the Seen and Unseen validation splits of the REVERIE Qi et al. (2020), a popular dataset to conduct our experiments. Cross-comparing results between these splits helps us infer the generalizability of our approach, while also enabling similar comparisons on other supervised learning approaches on this dataset.

REVERIE contains several human-annotated instructions for paths taken from the Matterport3D Chang et al. (2017) environment. The language used in the instructions uniquely capture various aspects of the environment. Each path is described as a discrete set of adjacent photorealistic panoramic images. They contain around 8 adjacent pano images (or hops) on average, which have been annotated with coarse-grained language guidance to reach target locations.

### 3.1 INSTRUCTION BREAKDOWN

We first look at breaking down coarse-grained guidance into *keyphrases*. Our objective is for the keyphrases to contain intermediate goals for the agent to use for sequential decision-making.

Consider the instruction:

On the second level go the bathroom inside the second bedroom on the right
***and***
replace the towels on the towel rack with the clean towels from the linen closet.

Observe that the conjunction ***and*** separates the *Navigational Component (NC)* of the instruction from the *Activity Component (AC)*. As such, a coarse-grained instruction can be broken down as,

*Coarse Instruction (I) = Navigation Component (NC) + Activity Component (AC)*

We empirically observe that this sentence form holds true with most instructions in REVERIE. The NC tells the agent about how to get to the target location, while the AC tells it what to do once it has reached there. In our case, since our objective is solely navigation, we focus on breaking up the NC into keyphrases and use the AC as a success detector for when the agent must stop.

One way to obtain keyphrases is to use *prepositions* in the instruction to break it up. Using this form of retrieval on the example above gives us:

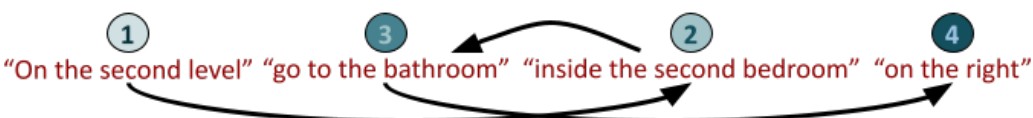

While this approach is simplistic, and breaks down instructions into rudimentary keyphrases for grounding, it often fails to capture the temporal nature of the instruction. Notice that "*go to the bathroom*" comes before "*inside the second bedroom*", when the agent needs to enter the second bedroom before going to the bathroom.

Alternatively, we look at using a Large Language Model to break it down for us. Using GPT-3 Brown et al. (2020) with the phrase "Break this down into steps." gives:

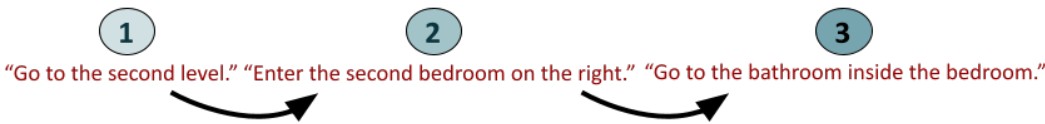

The instruction breakdown while being far more articulate, and is able to capture the sequential timeline i.e., "Enter the second bedroom" is before "Go to the bathroom".

## 3.2 LANGUAGE GROUNDING USING CLIP

After breaking down instructions into keyphrases, we use them to navigate inside a Matterport3D environment. Each node in Matterport3D is a panoramic image covering a 360 degree space around the agent. In order to select a direction for navigation within this panorama, we split it into 4 separate images. Each of these images covers approximately a 90 degree space around the agent, at a uniform horizontal elevation.

We use CLIP to ground keyphrases from both the Navigational (NC) and Activity components (AC) obtained in the instruction breakdown stage, and obtain two scores - a **Keyphrase Grounding Score (KGS)** and an AC grounding score. KGS helps the agent make sequential navigational decisions, while AC grounding score helps identify if the agent has reached the target location.

At each timestep, the image with the highest KGS is selected as the *CLIP-chosen image*, and is used to drive our navigational scheme. The AC grounding scores are for target object grounding, which gives us threshold or '*Stop Condition*' for when the agent needs to stop navigating.

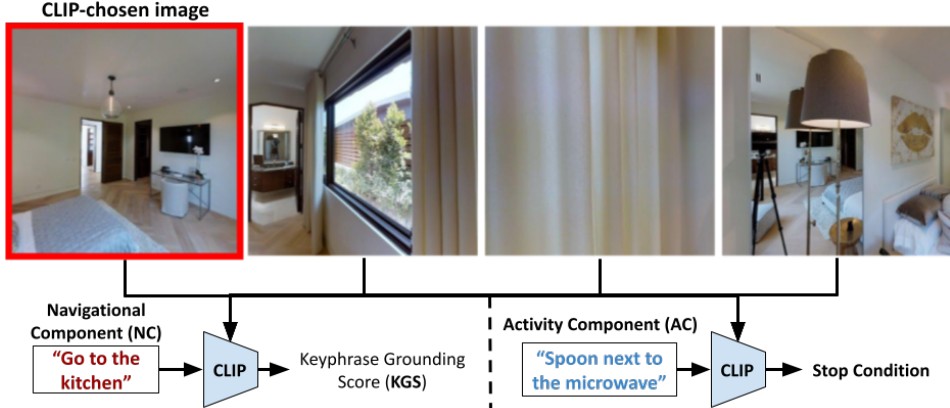

Figure 4: **CLIP Grounding** - We ground the Navigational Component (NC) on all the split images to obtain Keyphrase Grounding Scores (KGS). The *"CLIP-chosen image"* (highlighted in red) represents the one with the highest KGS, which drives our navigation algorithms. We also simultaneously ground the AC, and use the grounding score to determine if the agent has reached the target location—our *"Stop Condition"*.

The red image in Figure 4 represents the *CLIP-chosen* image for the given instruction. We observe that changing the instruction by adding more information to it also changes the CLIP-chosen image. For example, if the instruction here were "Go into to the kitchen that is next to the balcony on the second level" instead, the leftmost image is no longer the one with the highest grounding score. CLIP is sensitive to the language given to it for grounding, and this can hinder navigation performance. As such, we limit the words in the extracted keyphrases before computing CLIP scores.

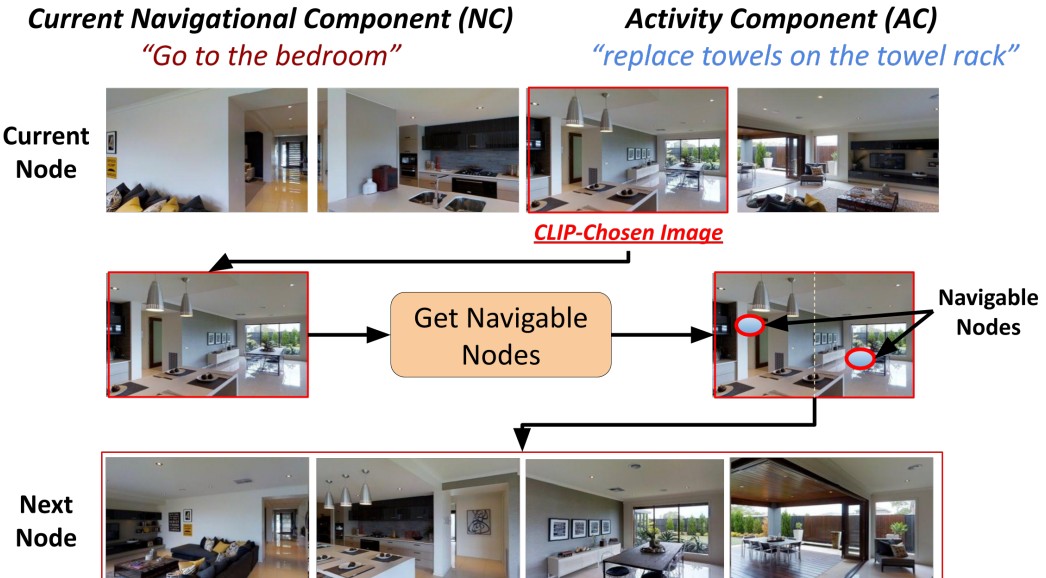

Figure 5: **CLIP-Nav** - We present a novel approach for zero-shot VLN, that utilizes CLIP to make sequential navigational decisions. At each timestep, a *CLIP-Chosen Image* is determined by grounding the current NC to each of the panoramic splits. The chosen image represents the direction our model has chosen for zero-shot navigation. In this case, it refers to the *bedroom* potentially being somewhere in the chosen direction. The AC grounding score gives us a stopping threshold for when to our agent believes it has reached the target. CLIP-Nav runs iteratively until this threshold is reached.

## 3.3 ZERO-SHOT NAVIGATION

The "*CLIP-chosen image*" obtained drives both our zero-shot navigation strategies.

### 3.3.1 CLIP-NAV

Figure 5 shows the overview of CLIP-Nav. At each time step, we split the panorama into 4 images, and obtain the *CLIP-chosen* image as explained in section 3.2. We obtain adjacent navigable nodes visible from this image using the Matterport Simulator, and choose the closest node. This is done iteratively till the '*Stop Condition*' described in the previous section is reached.

Beyond extracting *CLIP-chosen* images, the NC grounding score also determines when to select the next keyphrase. For instance, for "Go to the kitchen", if the grounding score is above a certain threshold, we assume that the agent has successfully navigated to the kitchen, and needs to execute the next keyphrase - "next to the balcony". In this way, we utilize CLIP not just for choosing navigational directions, but also for determining when the agent has reached intermediate goal locations.

### 3.3.2 SEQ CLIP-NAV

In order to improve CLIP-Nav, we incorporate a simple backtracking mechanism. Figure 6 presents an overview of this method.

We aggregate Keyphrase Grounding Scores across a sequence of $n$ nodes, and average them out to obtain a *Sequence Grounding Score (SGS)* as follows,

$$\text{SGS} = \frac{\sum_n \text{KGS}_i}{n},$$

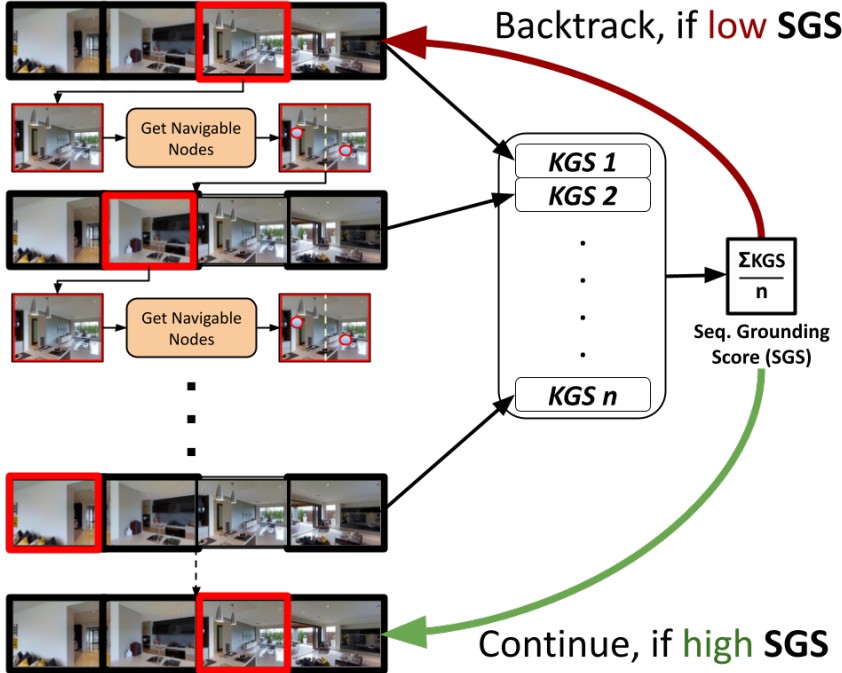

Figure 6: **Seq CLIP-Nav** - In order to improve the performance of CLIP-Nav, we incorporate a backtracking mechanism. CLIP scores across a sequence of timesteps are averaged to obtain a *Sequence Grounding Score (SGS)*. This score is then used to determine if the agent needs to go back a few nodes (backtrack) or not.

where KGS$_i$ is the KGS value of the CLIP-chosen image at at a particular timestep, and $n$ is the number of nodes. This score acts as a backtracking threshold to determine if the agent is heading in the right direction or needs to go back.

A low SGS value suggests that the agent might have gone down a path where it is not able to find the intermediate goal defined by the current NC. Alternatively, a high SGS shows that the agent is more confident of its ability in navigating the environment in a zero-shot manner.

## 4 RESULTS

Our results establish a baseline for zero-shot VLN, on the task of REVERIE. We use the following metrics to evaluate agent performance.

1. **Success Rate (SR)** - This is the fraction of episodes where the agent successfully reaches and stops at the target location.

2. **Success Weighted by Inverse Path Length (SPL)** - We use the definition provided by Batra et.al. in Batra et al. (2020). This is a standard metric used in several VLN tasks, and tells us about the optimality of the agent's success route, when compared with the ground-truth oracle path.

3. **Oracle Succss Rate (OSR)** - This is the fraction of episodes where the agent passes through the target location in its path, but does not stop there. This accounts for paths that may have overshot the target, due to inaccurate grounding.

4. **Relative Change in Success (RCS)** - We also compute the percentage relative change in performance between Seen and Unseen data, which gives us insight into the generalizability of each approach. This is defined as -

$$RCS = \frac{|Seen - Unseen|}{\max\{Seen, Unseen\}} \times 100$$

A lower score indicates that the agent is performing similarly across the splits, while a higher score indicates overfitting on the seen training environments.

| | Approach | SR(%) ↑ | | | OSR (%) ↑ | | | SPL ↑ | | |
|---|---|---|---|---|---|---|---|---|---|---|
| | | *Seen* | *Unseen* | RCS ↓ | *Seen* | *Unseen* | RCS ↓ | *Seen* | *Unseen* | RCS ↓ |
| **ZS** | Random Walk | 3.99 | 5.19 | 23.12 | 8.92 | 11.93 | 25.23 | 0.006 | 0.043 | 86.04 |
| | CLIP-Nav | 4.56 | 5.79 | 21.89 | 17.53 | 27.63 | 36.55 | 0.152 | 0.248 | **38.70** |
| | Seq CLIP-Nav | 12.34 | 14.97 | **17.56** | 19.47 | 24.46 | **20.40** | 0.212 | **0.450** | 52.88 |
| **SV** | FAST-MATTN | 50.53 | 14.40 | 71.50 | 55.17 | 28.20 | 50.69 | 0.455 | 0.072 | 84.17 |
| | AirBERT | 47.01 | 27.89 | 40.67 | 48.98 | 34.51 | 29.54 | 0.423 | 0.218 | 46.46 |
| | DUET | **71.75** | **46.98** | 34.52 | **73.86** | **51.07** | 30.85 | **0.639** | 0.337 | 47.26 |

Table 1: **Zero-Shot (ZS)** and **Supervised (SV)** results on the *Seen* and *Unseen Val* splits of REVERIE. The lowest RCS scores across all metrics indicate that our methods generalize significantly better in new environments over supervised approaches. On the unseen split, Seq. CLIP-Nav gives us the SOTA SPL result, and also improves upon the REVERIE baseline in terms of SR.

Table 1 compares our Zero-Shot (ZS) results on the Seen and Unseen Val splits of REVERIE with SOTA fully supervised methods. We also compare with a Random Walk approach that chooses a random neighboring node for 8 steps. This is the upper limit on the average path length of the dataset which is between 5-8 steps. Observe the improvement in performance of Seq CLIP-Nav over the REVERIE Baseline approach (FAST-MATTN), in terms of SR and SPL on the Unseen split. The Unseen SPL even outperforms the SOTA supervised learning approaches - Airbert and DUET, showing that when our agent does take the right path in an unseen environment, it tends to do so in a more optimal manner. Also notice the significant improvement in the SR performance of Seq CLIP-Nav over CLIP-Nav, even while the OSRs are similar. This indicates the strong influence of backtracking, in preventing the agent from overshooting once it reaches a target location.

On the Seen split, the supervised approaches outperform our methods in terms of SR and SPL, and this is quite evident since they have been trained on these environments. The higher RCS scores on these methods however indicate that they perform much worse on unseen environments in comparison to seen ones. This shows poor generalizability, and that they might have overfit on the training dataset. In contrast, the lower RCS scores across SR, OSR and SPL on our CLIP-based approaches indicates a far more consistent performance. We can thus quantitatively infer that our approaches generalize better in new environments over supervised approaches, satisfying one of our primary objectives for zero-shot navigation. This is a promising result, as it shows that our CLIP-based models without any finetuning are able to make consistent embodied navigational decisions irrespective of the type of environment that the agent is placed in.

Additionally, we also obtain "Best OSR" results of $47.51\%$ on CLIP-Nav, and $48.41\%$ on Seq CLIP-Nav on the unseen split. These are the single best values across all the house scans. The higher values when compared to the overall OSR shows that our approach is able to perform particularly well in certain types of environments, and analyzing the influencing visuo-lingual factors to transfer knowledge is part of a future study.

## 5 CONCLUSION AND FUTURE SCOPE

We tackle the problem of Vision-and-Language Navigation (VLN) in a fully zero-shot manner to address the issue of generalizability in unseen environments. Our approaches **CLIP-Nav**, and **Seq CLIP-Nav** give a far more consistent performance over fully supervised SOTA approaches on the REVERIE dataset when assessed by our generalizability metric — Relative Change in Success (RCS). **Seq CLIP-Nav** in particular gives improved results over the REVERIE baseline on the unseen validation set in terms of SR, and also improves upon SOTA supervised learning approaches in terms of SPL. This showcases the capability of CLIP without any dataset-specific finetuning in being able to make accurate sequential navigational decisions necessary for zero-shot VLN.

In the future, we intend to study the cross-dataset performance of CLIP-Nav on other fine and coarse-grained instruction sets in indoor and outdoor settings as well as when there is a cooperative dialog for instruction following Padmakumar et al. (2022); Gao et al. (2022). Our current backtracking score while simplistic, still drastically influences performance; improving this via meta-learning is a future study. Yet another exciting direction is a Virtual Reality (VR) experiment to learn human patterns in zero-shot instruction following. Analyzing people's zero-shot paths in VR environments could give us insight into their intuition for sequential decision making, and can be modeled to improve CLIP-Nav.

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
