# OpenReview forum: "CLIP-Nav: Using CLIP for Zero-Shot Vision-and-Language Navigation"
_robot-learning.org/CoRL/2022/Workshop/LangRob — LangRob 2022 Poster_

### Official Review · Reviewer_VyQ6 · 2022-11-12
**Well presented work using CLIP for zero-shot visual language navigation**

**Rating:** 8
**Confidence:** 3

**Review:**

**Summary**:

This work introduces CLIP-Nav, a method for zero-shot Visual-Language Navigation. Starting with the strong motivation that popular methods suffer large performance drops when evaluated on unseen environments, the work proposes utilizing CLIP to improve zero-shot performance when generalizing to new environments. To utilize CLIP, the method has three steps: instruction breakdown to elicit important keyphrases, vision-language grounding using keyphrases to identify salient locations and actions, and using the CLIP-chosen image to produce a navigation strategy. They demonstrate results in the REVERIE simulated navigation environment.


**Strengths**:
- The motivation on the importance of the zero-shot in the wild setting is helpful
- The paper is well presented with helpful visual explanations of the method
- The RCS metric is a strong indicator that CLIP-Nav is able to generalize to unseen environments

**Weaknesses**:
- There are quite a lot of notation and empirical metrics introduced; as someone not very familiar with visual navigation, it’s unclear to me how many of these are utilized by other works in the field and if there are more standard metrics or benchmarks you could compare to. Regardless, the presentation of Section 4 could be improved to make the results easier to understand
- Is there something special about CLIP particularly that lends itself to CLIP-Nav? It would be interesting to consider other visual grounding methods as well.

---

### Official Review · Reviewer_EyV1 · 2022-11-13
**Great work!**

**Rating:** 7
**Confidence:** 5

**Review:**

"Zero-shot" vision-and-language navigation has seen a lot of interesting progress in the recent months, and this paper presents yet another exciting way to use CLIP-like methods downstream. I am particularly happy with how the authors break down the instruction into subgoals that is beyond landmark recognition; that said, it is still a very strong bias to assume that simple subgoals would exist and we will continue seeing methods that uplift these assumptions.

Some thoughts:

- I found the "Activity Component" very hard to understand from text and a bit more description would be helpful. From the figure, it appears that it is also used for determining the stop condition, which resembles prior work in obtaining "success detectors" and "reward detectors" from images, and it would be useful to have a discussion of that.
- While it may not strictly be prior work, based on when the paper is accepted/published, I strongly recommend to at least discuss the differences in assumptions and capabilities wrt other up-and-coming zero-shot VLN papers, and maybe even compare to the relevant ones. Some that come to mind: [NL-Maps](https://arxiv.org/pdf/2209.09874.pdf), [VL-Maps](https://arxiv.org/abs/2210.05714), [CLIP-Fields](https://arxiv.org/abs/2210.05663), [LM-Nav](https://arxiv.org/abs/2207.04429) and so on.

---

### Decision · Program_Chairs · 2022-11-15

Accept (Poster)